# Mucopolysaccharidosis Type IIIE: A Real Human Disease or a Diagnostic Pitfall?

**DOI:** 10.3390/diagnostics14161734

**Published:** 2024-08-09

**Authors:** Karolina Wiśniewska, Jakub Wolski, Magdalena Żabińska, Aneta Szulc, Lidia Gaffke, Karolina Pierzynowska, Grzegorz Węgrzyn

**Affiliations:** 1Department of Molecular Biology, Faculty of Biology, University of Gdansk, Wita Stwosza 59, 80-308 Gdansk, Poland; karolina.wisniewska@phdstud.ug.edu.pl (K.W.); magdalena.zabinska@phdstud.ug.edu.pl (M.Ż.); aneta.szulc@phdstud.ug.edu.pl (A.S.); lidia.gaffke@ug.edu.pl (L.G.); karolina.pierzynowska@ug.edu.pl (K.P.); 2Psychiatry Ward, 7th Navy Hospital in Gdansk, Polanki 117, 80-305 Gdansk, Poland; jakub.wolski@o2.pl

**Keywords:** mucopolysaccharidosis, Sanfilippo disease, MPS IIIE, classification, *ARSG* variants

## Abstract

Mucopolysaccharidoses (MPS) comprise a group of 12 metabolic disorders where defects in specific enzyme activities lead to the accumulation of glycosaminoglycans (GAGs) within lysosomes. This classification expands to 13 when considering MPS IIIE. This type of MPS, associated with pathogenic variants in the *ARSG* gene, has thus far been described only in the context of animal models. However, pathogenic variants in this gene also occur in humans, but are linked to a different disorder, Usher syndrome (USH) type IV, which is sparking increasing debate. This paper gathers, discusses, and summarizes arguments both for and against classifying dysfunctions of arylsulfatase G (due to pathogenic variants in the *ARSG* gene) in humans as another subtype of MPS, called MPS IIIE. Specific difficulties in diagnostics and the classification of some inherited metabolic diseases are also highlighted and discussed.

## 1. Introduction

Mucopolysaccharidoses (MPS) are a heterogeneous group of genetically determined metabolic disorders, belonging to lysosomal storage diseases (LSD) with complex pathomechanisms [1]. The primary cause of the disease is a disturbance in the activity of lysosomal enzymes responsible for the hydrolysis of glycosaminoglycans (GAGs)-long, linear polysaccharides that are a multifunctional group of compounds due to their polyanionic nature. Incompletely degraded GAGs accumulate in lysosomes, thereby disrupting normal cell functions and subsequently causing tissue-level abnormalities and organ dysfunctions [1,2,3]. There are currently 12 ‘classical’ (conventional, i.e., those in which GAG accumulation arises from a severe decrease in an activity of an enzyme directly involved in GAG degradation; not counting MPS-plus syndrome (MPSPS), where GAG storage is caused by another mechanism) types and subtypes of MPS, where the classification is based on both the type(s) of accumulated GAG(s) and the enzyme affected by the defect. In addition, depending on the absence/presence of symptoms of the nervous system disorders, MPS can be subdivided into non-neuronopathic types (MPS IVA and IVB, VI, VII and X) and neuronopathic types (MPS I, II, all subtypes of MPS III and VII) [4].

Recently, several papers have mentioned the 13th type of MPS, MPS IIIE [5,6,7,8,9]. Until recently, MPS IIIE was described as a subtype found only in animal models [10]. The gene variant causing this MPS has been detected in humans relatively recently and has been described as Usher disease (USH) type IV [11,12,13]. The aim of this study is to discuss the arguments both for and against the recognition of MPS IIIE in humans.

## 2. Mucopolysaccharidosis

As mentioned earlier, MPS is a heterogeneous group of diseases in which symptoms can vary not only between types but also within subtypes [6] (see Table 1 for details). Nevertheless, some symptoms, like coarse facial features, dysostosis multiplex, hepatosplenomegaly, cardiovascular disorders, cognitive impairment (in neuronopathic forms), recurrent respiratory infections, and diarrhea, occur in the majority of patients, though with various severities [14,15]. 

Disorders that develop in the course of MPS, particularly those affecting the osteoarticular and central nervous systems, are irreversible after reaching a specific threshold, thus, early diagnosis and treatment are extremely important [16,17]. However, the diagnosis of MPS is difficult for a number of reasons. MPS is a multisystemic, rare disease with a progressive nature [3,15]. In practice, this means that in a child who appears healthy at birth, the first symptoms appear over time and may not be obvious. As the disease progresses, the abnormalities become more and more apparent, at the same time that further disorders appear which may be regarded as symptoms of another disease [16,18]. An additional difficulty is the heterogeneity of symptoms among patients and the rarity of MPS. This results in a delayed diagnosis at best (often preceded by a misdiagnosis of another disease), while at worst, a lack of diagnosis [19]. The diagnosis of MPS is often based on the clinical picture, and sometime a patient’s history (possibility of MPS in relatives). When MPS is suspected, the first test performed is usually the assessment of the urinary GAG level determination. The next step is assessing the activities of lysosomal enzymes, only the result of which allows a diagnosis of MPS to be made. Molecular tests are then performed to identify gene variants to assess the progression of the disease and plan appropriate treatment [19,20].

Sanfilippo disease is a type III of MPS in which only one GAG, heparan sulfate (HS), accumulates. Because the HS degradation pathway involves several enzymes, subtypes of Sanfilippo disease are distinguished, where the current classification includes four of them (subtypes A–D) [21]. Patients with MPS III, like those with other MPS types, show various symptoms, i.e., facial dysmorphia, hepatosplenomegaly, osteoarticular abnormalities, etc., but the somatic disturbances are relatively mild, relative to those with other MPS types. In contrast, the central nervous system (CNS) disorders are especially severe [19]. 

In the course of MPS III, children initially develop normally until around 2–3 years of age, then the first symptoms begin to appear. The slowing down or stopping of normal development and a gradual loss of acquired skills are observed [9,22]. Features of facial dysmorphia begin to become apparent. Symptoms such as frequent ear and upper respiratory tract infections or diarrhea are often underestimated. Around 3–4 years of age, cognitive functions deteriorate significantly, and sleep disorders and behavioral problems appear (including aggressive-like behavior, impulsivity, anxiety disorders, and/or autistic disorders) [23,24]. Skeletal abnormalities become more severe and more apparent. In the teenage years, a child’s quiescence, cognitive decline, and dementia are observed. Patients lose the ability to function and move independently. In milder forms, the symptoms of the disease usually appear later and are not as pronounced as in severe forms [19]. 

Animal studies have shown the existence of another enzyme involved in HS degradation, arylsulfatase G (ARSG) [10,25]. The results of experiments with *Arsg*-knockout mice showed a similarity to mouse models of the other four types of MPS III. The *Arsg* dysfunction and the deficiency of the corresponding enzyme led to HS accumulation in lysosomes [10]. HS storage was evident in both CNS and somatic organs, like the liver and kidney [10,25,26]. Secondary accumulations of complex lipids (glycolipids, gangliosides, lipofuscins, cholesterol) and subunit c of mitochondrial ATP synthase (SCMAS) were also observed. Some behavioral abnormalities, typical for animal models of the other 4 MPS III subtypes, were also indicated [10]. Since a pathogenic variant in the *ARSG* gene in a homozygous configuration has also been identified in humans [11,12,13], while not causing GAG accumulation, the question arises as to whether the classification of MPS should be expanded to include another subtype?

## 3. MPS IIIE in Humans—Whether It Is or Not?

The correct diagnosis of a disorder, even when effective treatment options are not available, is always important, at least to provide the best possible management and care to patients. From the perspective of parents/guardians and the patient, knowing and understanding the cause of the disorder also allows for a kind of psychological comfort. By knowing the cause, they gain an insight into the illness, which reduces feelings of helplessness, enables them to get used to the situation and gives them a sense of control. From a medical point of view, a correct diagnosis, even in the absence of drugs to cure the patient, allows optimal care to be planned. By knowing the disease, it is possible to more or less anticipate its course and introduce preventive measures for possible complications. It is also possible to plan and implement symptomatic treatment, seek experimental therapy programs, and, above all, prevent the use of treatment methods that may be not ineffective or even harmful. Therefore, the discussion regarding the possible occurrence of MPS IIIE in humans is an important issue requiring attention.

### 3.1. Why Yes?

#### 3.1.1. ARSG Activity

In 2010, Abitbol et al. published a paper describing adult American Staffordshire terriers (ASTs) as a canine model of neuronal ceroid lipofuscinoses (NCLs), a lysosomal storage disease with neurodegeneration and neurological disorders, caused by the accumulation of autofluorescent lipofuscin and fatty lipopigment [27,28]). Symptoms of the disease begin to be seen in the juvenile period, very rarely only in adulthood. At the time of publication of that paper [27], the cause of inherited forms of CLN4 was not known. It was observed that ASTs sometimes develop symptoms similar to those characteristic of CLN4, i.e., locomotor ataxia, absence of visual impairment, marked cerebellar atrophy, and accumulation of a specific lipopigment in Purkinje cells and thalamic neurons (revealed abnormal lysosomes filled with inclusions). In addition, the disorders appeared late and developed slowly [27]. Studies have shown the presence of variants in the *ARSG* gene, leading to a decrease in the activity (by as much as 75%) of the *N*-sulfoglucosamine-3-*O*-sulfatase (arylsulfatase G), an enzyme encoded by this gene. Variants in this gene have been proposed as a possible cause of NCLs with onset in adulthood [27]. However, the exact function of the enzyme was first described in 2012 by Kowalewski et al. [10]. In order to determine the function of the enzyme, the researchers created the knock-down (KO) mouse mutant in the *Arsg* gene. The results of that study showed that ARSG is one of the lysosomal hydrolases involved in GAG degradation [25]. Loss of activity of this enzyme led to HS accumulation in mice due to the inability to remove 3-*O*-sulfated *N*-sulfoglucosamine residues of heparan sulfate (a simplified diagram of HS degrading enzymes, including ARSG activity, is shown in Figure 1) [10]. Not only did the KO mouse model of ARSG deficiency show features similar to MPS III (described in the next section), but also the accumulation of autofluorescent material, which was observed in *ARSG*-deficient dogs [25,27], was observed. In view of the results described above, a link between *ARSG* variants and the pathogenesis of MPS was proposed and it was suggested that the previously described canine model of NCLs may also be an animal model of MPS III [10]. It is now indeed considered that the variant in the *ARSG* gene, previously described in the canine model, is associated with the pathogenesis of MPS, at least in animals (specifically, in dogs and mice) [29,30].

#### 3.1.2. Mild/Attenuated Type of MPS

As mentioned earlier, in 2012, Kowalewski et al. published a paper describing the phenotype of the constructed *Arsg*-knockout mouse line (ARSG KO) [10]. This model showed similarities to other mouse models of MPS III in terms of both biochemical and behavioral abnormalities. HS accumulation was observed in the liver and brain, the organs typically affected in MPS III, but without features of organomegaly. Accumulation of GAG was also observed in organs [10,31]. In addition, significant enlargement of lysosomes was observed in the cells of some brain areas. A secondary accumulation of gangliosides and other lipids has also been observed, as also occurs in the course of other MPS types [10,32]. As the mice aged, difficulties in acquiring new skills have been observed [25]. Stenosis (SCMAS) has been documented in MPS III and has also been confirmed in the ARSG KO mice [25,33,34]. A common feature of several lysosomal storage disorders, including MPS, is retinal degeneration, which was the earliest observed disorder in the ARSG KO mice studied [26,35,36]. In short, mice with the dysfunctional *Arsg* gene (in the homozygous state) show some characteristics of LSD, and especially MPS III.

The aforementioned ARSG KO model has features characteristic of MPS; however, compared to other MPS III models, they appear to be relatively mild [25,26]. Nevertheless, it is not a very unusual situation, as, for example, MPS I is subdivided into clinical subtypes (from extremely severe, through moderate, to mild; called also Hurler, Hurler/Scheie, and Scheie disease, respectively) not because of the different molecular basis, but because of differences in the severity of the symptoms [15]. Similarly, MPS II, often very severe in its course, can develop into the attenuated type, although the variant in both cases involves the same gene [37]. The severity of the symptoms, in both cases (MPS I and II), depends on the HS concentration and the ratio between different kinds of GAGs [38]. Although the subdivision of MPS III into subtypes is due to differences in molecular basis (mutant gene and deficient gene product–an enzyme), it is not obvious why subtypes IIIA and IIIB are known to be more severe in their courses than subtypes IIIC and IIID [39]. It also happens that people with the theoretically more severe MPS III subtypes (A and B) remain undiagnosed for years because they have developed mild symptoms (with minor cognitive impairment or even non-neuronopathic). An interesting series of 12 patients with MPS III was described, with adult-onset phenotypes and mild cognitive impairment or even non-neuronopathic phenotypes [40]. In six of these patients, the main symptom (in addition to decreased enzyme activity and elevated urinary GAG levels) was retinal dystrophy (RD)/RD with visual impairment [40]. In four more patients, RD developed after diagnosis. It should be mentioned that the oldest patients were 50–70 years old (7 patients) and three were over 40 years old, while the life expectancy of patients with classic MPS III is about 20–30 years [19,40]. 

De Falco et al. described the case of a patient who was diagnosed with Usher’s disease (USH) at around 30 years of age due to retinitis pigmentosa (RP) and sensorineural hearing loss (SNHL) [41]. The patient’s condition deteriorated over the next 20 years. At the age of 53 years, the patient was re-diagnosed. No behavioral, cognitive or memory disorders were detected. Physical examination did not reveal features of facial dysmorphia or organomegaly [41]. On neurological examination, no specific abnormalities were detected except for minimal cerebellar features, including mild telekinetic tremor (left > right) and dysdiadochokinesia [41]. Attention could be drawn to the short stature of the male patient (150 cm). Exome sequencing did not reveal variants conditioning USH, but variants in the *SGSH* gene, causing MPS IIIA, were detected. GAG measurement and enzyme activity tests showed elevated HS levels and no detectable SGSH activity. Thus, the final diagnosis of MPS IIIA was made [41]. This case indicates that Sanfilippo disease may present with atypical symptoms, but two crucial features should be evident, i.e., HS storage and deficiency in one of enzymes responsible for GAG degradation.

The above statement is supported by the facts that attenuated forms of MPS may run their course without the characteristic dysmorphic facial features, and patients’ life expectancies may not differ from those of healthy individuals [40,42]. The main symptoms may not necessarily be retardation/stunting, cognitive impairment, or hepatosplenomegaly, but they can include visual impairment, skeletal problems without inflammatory features, and hearing and cardiac problems [42]. Moreover, urinary GAG levels may be normal [42]. Therefore, the ARSG KO mouse model may indeed be a model of MPS III, whose phenotype is benign.

### 3.2. Why Not?

#### 3.2.1. The Animal Model and Humans

Doubts about the validity of the MPS diagnosis in the cases of variants in the *ARSG* gene may already arise at the animal model level. Admittedly, studies in the mouse model demonstrated a number of disorders converging (and key) to those observed in animal models of other MPS III subtypes, i.e., high levels of HS in lysosomes, secondary lipid accumulation, liver involvement, or CNS abnormalities [10,25,26]. However, some differences emerged. First of all, it is important to note that the mouse model described by Kowalewski et al. is of the knockout type, i.e., it is completely devoid of ARSG activity, thus a severe phenotype and a rapid course could be expected [10,43,44,45,46]. Kruszweski et al. pointed to a relatively late onset of symptoms compared to mouse models of other MPS III types, and the authors of the model themselves indicated a milder phenotype compared to other MPS III subtypes [25,26]. The mice did not show symptoms of the disease until the age of 12 months, despite HS storage in CNS and peripheral organs [10]. The major neurological finding of ARSG KO was ataxia with massive Purkinje cell degradation; however, the earliest sign was retinal degradation occurring between 1–6 months of age in mice, even before the onset of neurological abnormalities, while changes in the brain structure were rather limited to the cerebellum [26]. Retinal degradation also occurs earlier in mouse models of MPS III, but the earliest signs of neurological disorders are primarily motor and behavioral abnormalities, such as hyperactivity or aggression [43,44,45,46]. Although behavioral and cognitive abnormalities were observed in the ARSG KO mice, they appear significantly later and are not as severe as in other MPS III mouse models [43,44,45,46]. Moreover, motor dysfunction was not observed in the ARSG KO mouse model [25]. Pathological changes in brain structures are much more extensive (hippocampus, cortex, cerebellum, spinal cord, microglia, astroglia) in other MPS III subtypes than in the *Arsg* mouse mutants [44,45,46,47]. 

In addition to HS accumulation, dermatan sulfate (DS) accumulation has also been observed in the mouse *Args* knockout model [25]. It is worth noting that in MPS II, both HS and DS accumulate [38]. Clinically, MPS II is divided into severe and attenuated forms [37], which could account for the relatively milder course, compared to MPS III [25]. Furthermore, the aforementioned accumulation of SCMAS proteins as well as retinal degradation are observed in MPS II [34,35]. In fact, ARSG KO mice resemble, to some extent, a mild form of MPS II, with a defined difference in molecular basis.

*Arsg*-knockout mice were created to study the action of ARSG [10]. By design, this model was not intended to reflect human disease, as at the time of its development, MPS IIIE has not been reported in humans [25,26]. In fact, mouse models of MPS III A–D largely correspond to the disorders observed in patients. The differences are quite minor (with some obvious limitations, like the impossibility of studying speech disorders in animals) and relate to the speed of onset of symptoms rather than the subtype of MPS III [43,44,45,46]. Although the mouse model of ARSG deficiency does indeed show disorders similar to those observed in mouse models of the other types of MPS, in humans it causes a completely different disorder, defined as USH type IV [13]. 

USH is a recessively-inherited disease with SNHL and RP, with or without vestibular dysfunction [48]. Based on the age of onset of symptoms, their severity and the speed with which the disease progresses, there are three main types of USH, namely I, II, and III [13]. In 2018, Katheb et al. described variants in the *ARSG* gene (a homozygous missense variant) as a cause of USH, describing it as an atypical USH phenotype [11]. No vestibular system involvement was observed in any of the subjects, SNHL and RP (the characteristic phenotype of ring-shaped retinal atrophy along the arcades) with late, for USH, onset were observed in all five patients. Investigations showed no neurological abnormalities, and magnetic resonance imaging showed no brain abnormalities, similar to abdominal ultrasound, which also showed no changes. Three patients showed skeletal changes (osteoporosis), but it should be noted that the patients were 50–72 years old at the time of the study [11]. It was tested whether the variant detected in the patients’ *ARSG* gene (D45Y) affects the activity of ARSG and other lysosomal enzymes. The results showed a decrease in ARSG activity without affecting other lysosomal hydrolases. At the same time, GAG levels remained at the upper limit of the norm [11]. It was acknowledged that the described patients’ phenotypically did not fit into the classical USH variants, but also did not show features typical of MPS. It was suggested that updating the USH classification be considered [11]. Since then, a total of 15 variants in the *ARSG* gene have been described, leading to the development of the disorder now recognized as USH type IV [11,12,13,49,50,51].

The diagnosis of USH type IV, rather than Sanfilippo disease, in the above described patients, was based on the fact that decreased ARSG activity did not correlate with other features typical of MPS, like lysosomal abnormalities, somatic abnormalities (like hepatosplenomegaly or osteoarticular disorders), and especially elevated GAG levels (which remained around the upper limit of normal values in these patients) [11,12,13]. Although hearing abnormalities are typical of many types of MPS, including MPS III, and visual abnormalities (corneal clouding, glaucoma, retinopathy, or optic nerve abnormalities) are characteristic of neuronopathic types of MPS [52,53], no CNS abnormalities typical of the neuronopathic types of MPS, including MPS III, were found in the described patients [11,12,13,45,46,47]. In 2021, Kowalewski et al. confirmed that the ARSG dysfunction is connected to USH, particularly untypical USH type IV [54]. Interestingly, it was suggested that other, as yet unidentified, 3-*O*-sulfatase(s) of HS may exist, as ARSG could not catalyze the reaction using a substrate which contains a free amino group [54]. Indeed, this might explain the discrepancy between the lack of ARSG function and normal GAG levels in patients diagnosed for USH type IV, as the auxiliary function of another, fully functional enzyme might compensate for ARSG deficiency. Nevertheless, the discovery of such sulfatase(s) is necessary to verify this hypothesis positively.

Interestingly, a similar situation applied to the classification of variants in the *ARSK* gene. There is a mouse model, where HS and DS accumulation occurs due to dysfunction of one of the lysosomal hydrolases, arylsulfatase K (ARSK) [19]. In 2020, Trabszo et al. published a paper on the ARSK-deficient mouse model, in which elevated levels of HS and DS were noted in the liver, spleen, and brain, with a concomitant decrease in ARSK enzymatic activity [8]. Although behavioral changes in ARSK-deficient mice were observed, they were considerably milder than those in models of other types of MPS. Additionally, features indicative of CNS abnormalities were absent. The effects of ARSK deficiency on the skeletal system (whose abnormalities are typical in MPS) were also not observed. The authors of that study proposed distinguishing MPS IIB, as a subtype of MPS [8]. However, MPS II (Hunter syndrome) is one of the neuronopathic types of MPS, in which iduronate 2-sulfatase (IDS) function is impaired, resulting in deposition in HS and DS in lysosomes [38]. Abnormalities characteristic for MPS II include skeletal abnormalities, organomegaly, facial dysmorphia, short stature, inguinal and umbilical hernias, hearing impairment/loss, and neurological abnormalities with functional impairment and corneal opacity [38]. Sometimes, however, patients with a benign phenotype do not develop CNS abnormalities [38]. One year later, Verheyen et al. published a paper in which a variant leading to ARSK deficiency was described in humans [55]. However, the observed disorders differed from those described in the mouse model devoid of the *Arsk* function. The patients developed osteoarticular disorders without behavioral abnormalities. Symptoms typical of MPS were observed, i.e., short stature, recurrent ear infections, sleep problems, mild visual system abnormalities, and facial dysmorphic features. MPS IV was initially suspected, but subsequent investigations (assessment of enzymatic activity) ruled out this possibility. Although urinary GAG levels were normal, liquid chromatography/mass spectrometry (LC-MS/MS) showed a significantly increased level of DS [55]. Therefore, a diagnosis of MPS type X was proposed [55]. Over time, additional patients with similar disorders and a similar diagnostic process began to be reported [56,57]. Skeletal abnormalities, no behavioral incapacity, or CNS abnormalities with typical symptoms suggestive of MPS were found. Initial assessment of urinary GAG levels showed no abnormalities, but repeated testing suggested MPS IV. Evaluation of enzymatic activity together with genetic testing and GAG examination with LC-MS/MS confirmed ARSK deficiency with accumulation of DS [56,57]. Like Verheyen et al., both Rustad et al. and Sun et al. classified the described cases as MPS X [50,51,52].

Animal models of human diseases are certainly much more advanced than cellular models, on which it is not possible to show/consider the bigger picture of how the organism as a whole works. However, an animal model is still only a model. Indeed, there are many examples of studies where potential therapeutics with promising results in animal studies have not worked in humans. The same principle applies to disorders observed in the course of a disease. Not all symptoms seen in humans can be transferred to animal models and, as is evident in some MPS types, this also works the other way round. 

#### 3.2.2. MPS-like Symptoms

Mucopolysaccharidosis-plus (MPSPS) is a disease in which variants in the *VPS33A* gene result in cellular accumulation of DS and HS (high urinary concentrations of both GAGs) while the activities of all the enzymes responsible for GAG degradation remain unaltered [58,59]. The clinical picture is remarkably similar to that seen in patients with ‘classic’ MPS, including skeletal abnormalities such as excessive joint stiffness or dysostosis multiplex, facial dysmorphic features, and cardiovascular abnormalities. In addition, as in the neuronopathic types of MPS, CNS abnormalities (psychomotor retardation and developmental delay) appear, which may be associated with HS accumulation [19,22,58]. Nevertheless, the variant in the gene that causes MPSPS represents a distinct disease entity within LSD [60]. No classification as benign/atypical MPS I or II has been proposed, as the molecular mechanisms of this disease are significantly different from those of ‘classic’ MPS disorders. 

#### 3.2.3. One Gene, Different Diseases

GM1 gangliosidosis and MPS IVB are LSDs in which multiple disorders are caused by variants in the *GLB1* gene which encodes acid β-galactosidase [16]. As a result, keratan sulphate (KS) and ganglioside GM1 accumulate in cells [16,61]. Both GM1 gangliosidosis and MPS IVB present with skeletal abnormalities, facial dysmorphic features, hepatosplenomegaly, visual disturbances, and frequent diarrhea (symptoms typical of MPS) [62,63]. However, GM1 gangliosidosis is a sphingolipidosis, a neurodegenerative disease in which the accumulation of primarily GM1 and GA1 gangliosides occurs in the nervous system [62], while MPS IVB is characterized by predominant KS accumulation in bones and cartilage with little or no CNS involvement (if present, CNS dysfunctions are relatively mild and develop secondarily to the main disorders) [19]. Although these two diseases are remarkably similar, not only in terms of genetic background, but also the stored materials and the observed disorders, MPS IVB is still classified as a mucopolysaccharidosis rather than another subtype of gangliosidosis. Conversely, no reclassification of GM1 gangliosidosis as another MPS type/subtype has been proposed so far. This raises the question of why a variant in the *ARSG* gene, which does not even give LSD-typical symptoms and is already classified as a specific disease entity, USH type IV, should be renamed as MPS IIIE?

RP is the most common type of inherited retinal degeneration with diverse genetic backgrounds [64]. At first, RP usually manifests with night blindness, followed by visual field loss, and eventually leads to total blindness [64]. RP can be a disease in itself or a component of other conditions such as USH, mitochondrial disorders, or inborn defects of metabolism, like MPS [64,65,66]. Haer-Wigman et al. described six patients with a variant in the *HGSNAT* gene, a defect of which leads to impaired heparan-α-glucosaminide-*N*-acetyltransferase (HGSANT) activity, HS accumulation, and a diagnosis of MPS IIIC [64]. In order to exclude the possibility of a benign type of MPS IIIC, patients underwent physical examinations and biochemical tests, which are standard procedures in the diagnosis of MPS. Physical examination did not reveal abnormalities typical of LSD/MPS. One patient was found to have mild hearing loss at 59 years of age, and none of the six patients showed CNS disorders or behavioral abnormalities. Examination of the HGSANT activity showed its significant decrease, compared to the reference value for healthy subjects, but the patients’ scores remained significantly higher than those of individuals diagnosed with MPS [64]. The final diagnosis of all six patients was non-syndromic RP [64]. In 2020, Schiff et al. conducted a study in a group of 20 patients with a variant in the *HGSNAT* gene who were diagnosed with RP [67]. Biallelic variants of the gene were detected in 17 patients, but none showed features typical of MPS IIIC. Enzyme testing from leukocytes showed a significant but mild reduction in the activity [67]. It is noteworthy that four of the identified variants were previously described as occurring in MPS IIIC [67]. 

Despite these ostensibly paradoxical cases, the fact that the same variant can lead to radically different phenotypes is not surprising. It is important to remember that when investigating the effect of single gene variants, one must take into account the broader genetic and environmental context. The ‘final result’ is influenced not only by the main variant, directly determining the development of the disease, but also by the genetic background, epigenetic factors, gene-gene interactions, and environmental factors [68]. All of these can affect differential penetrance and expression, and exacerbate/mitigate primary disorders [68]. RP resulting from variants in genes that normally lead to LSD also involves genes that condition neuronal ceroid lipofuscinoses (NCLs) [13]. Pathogenic variants in the *CLN3*, *CLN5*, and *CLN7* genes usually lead to accumulation of autofluorescent lipofuscin and fatty lipopigment in lysosomes of neuronal cells. This results in progressive neurological deterioration with dementia, epilepsy, loss of vision, motor disturbances, and early death [69]. However, several research groups described variants in the above-mentioned genes associated with the diagnosis of isolated retina degeneration [70,71,72,73]. A similar situation could apply to the *ARSG* gene variants described in USH type IV, but these are variants that lead to loss of function/activity and yet do not cause the typical LSD/MPS symptoms. Therefore, it is difficult to assume that other variants will be associated with a wider spectrum of symptoms or that symptoms will develop later [13]. It should be emphasized that the diagnosis of atypical/mild forms of MPS/NCLs was not proposed in the described cases. In contrast, the diagnosis was a completely different disease entity, RP.

#### 3.2.4. Diagnostics

The first step in the diagnosis of MPS is to suspect the presence of this disorder based on the patient’s clinical features and information obtained from the patient’s history [19]. Clinical features that may suggest a diagnostic workup for MPS include facial dysmorphia (thickened features, prominent forehead and lips), coarse hair, dental abnormalities, thoracolumbar kyphosis, joint stiffness, short stature, and concomitant CNS disorders (diagnosis for neuronopathic types of MPS) (details presented in Table 2) [18,19]. A family history of MPS should be of particular interest, but also frequent respiratory tract and ear infections, recurrent diarrhea, or problems with sight or hearing, or a history of hepato- or splenomegaly is also considered [15,18]. 

The next step is to perform a test to assess urinary GAG levels, but it is important to remember that the results of such a test are not the basis of confirmi7ng the diagnosis of MPS (if elevated GAG levels are found) or rejecting the diagnosis (if GAG levels remain normal) [3,74]. GAG levels may change with age, additionally they may not be elevated in individuals with a mild disease phenotype, and it is also the case that results may be false positives. Only the assessment of enzymatic activity, together with the clinical picture and history, allows the diagnosis of MPS to be made [75]. Additional investigations are helpful during diagnosis, including radiographs to evaluate joints and bones (especially hip joints and metacarpal bones), brain magnetic resonance imaging, or various biochemical tests [19,74]. Molecular analysis to detect pathogenic gene variant(s) can finally confirm the disease, while is usually performed after initial diagnosis. However, this also facilitates assessing the possible course of the disease and planning appropriate treatment [3,15,19]. A simplified diagnostic scheme for MPS is shown in Figure 2.

Examples of MPS misdiagnosis or diagnostic difficulties were published in the literature [19]. An interesting example is the paper by Langereis et al. who described a patient in whom diagnosis was undertaken for Fabre disease [76], an LSD in which, due to a deficiency in α-galactosidase A (α-Gal A) activity, there is an accumulation of glycosphingolipids [77]. A 32-year-old man was hospitalized due to ischemic stroke. However, because of the lack of predisposing factors, the diagnosis of Fabry disease was started. However, α-Gal A activity did not reveal abnormalities, while the test revealed a decreased activity of α-L-iduronidase, an enzyme whose defect leads to MPS I [76]. Thus, diagnosis for this MPS type was initiated. Despite the low activity of the enzyme (while still higher than in MPS patients), the urinary GAG content (both quantitatively and qualitatively) remained normal. Genetic analysis confirmed the presence of variants in the *IDUA* gene, suggesting MPS I, Scheie phenotype (the mild form of MPS I), but the diagnosis was not confirmed [76]. Until the incident from which the man returned to full function and for which he was hospitalized, the patient remained healthy. He negated any joint mobility problems, and his family history also did not indicate a possible genetic disorder burden associated with metabolic diseases [76]. The patient’s height (180 cm) also did not indicate typical abnormalities for MPS. No organomegaly or dysmorphic features were found on examination. Ultimately, the suspicion of mild MPS type I was rejected [76].

## 4. Conclusions

There are arguments for and against the classification of the disease caused by pathogenic variants in the *ARSG* gene as MPS IIIE in humans. They are summarized in Table 3. Nevertheless, since both GAG storage and deficiency in activity of an enzyme involved in GAG degradation are required to make an MPS diagnosis, in our opinion, despite mice with *Arsg* dysfunction resembling those used as models in other subtypes of MPS III, classification of the corresponding disease in humans as MPS IIIE should be considered only if patients are described with *ARSG* variants as the only genetic defect, while developing biochemical features and symptoms characteristic of MPS. Otherwise, we suggest that there is no reason to include MPS IIIE in the list of human MPS types/subtypes.

## Figures and Tables

**Figure 1 diagnostics-14-01734-f001:**
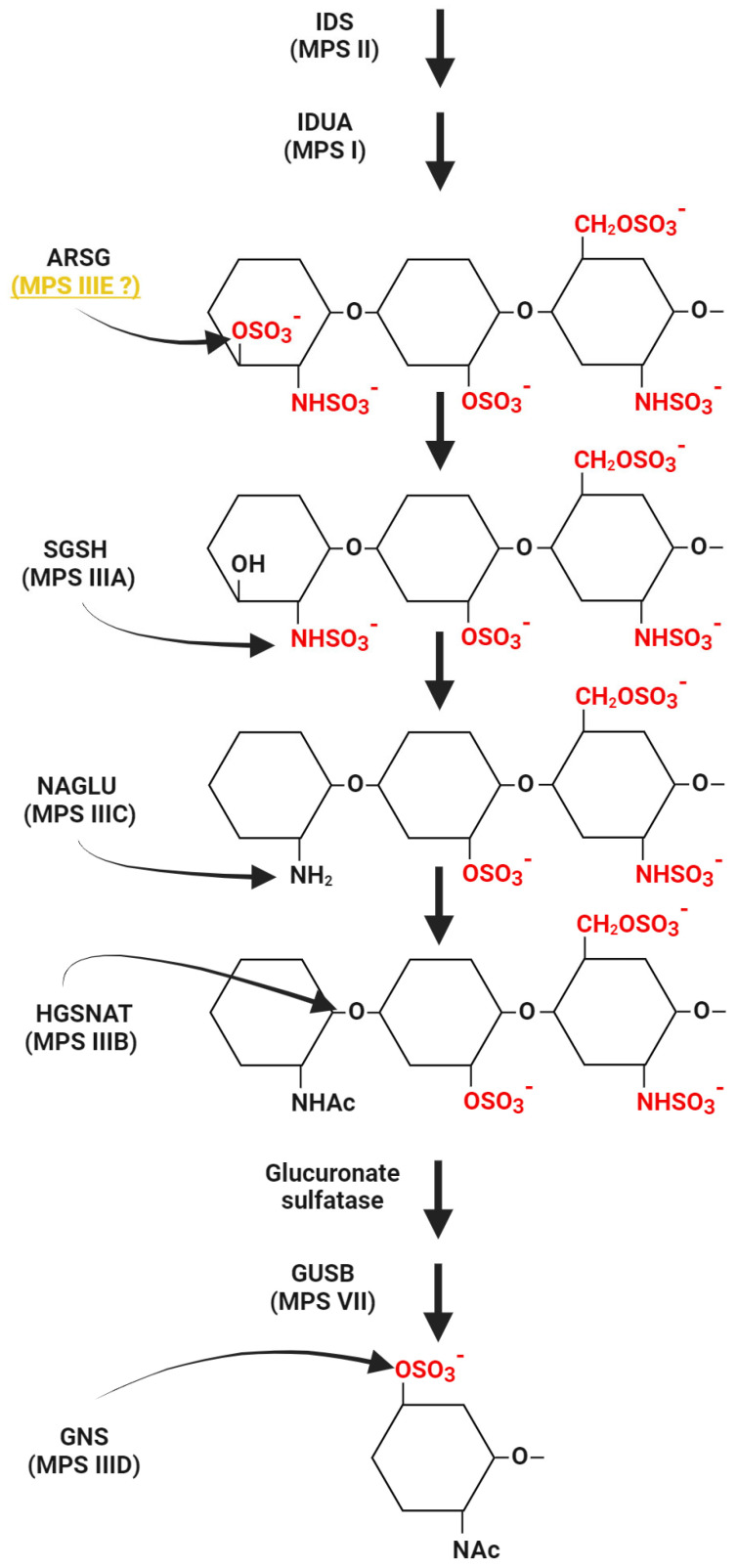
The pathway of HS degradation with a special emphasis on enzymes for which dysfunctions cause specific subtypes of MPS III. Reactions catalyzed by these enzymes are indicated, while other steps and enzymes for which deficiencies are responsible for other MPS types are shown in a simplified form. Whether deficiency of arylsulfatase G (ARSG) causes MPS IIIE in humans or not is a disputable issue; thus, this MPS subtype is followed by a question mark. Abbreviations: IDS, iduronate-2-sulfatase; IDUA, α-L-iduronidase; ARSG, arylsulfatase G; SGSH, Heparan-*N*-sulfatase; NAGLU, α-*N*-acetylglucosaminidase; HGSNAT, heparan α-glucosaminide-*N*-acetyltransferase; GUSB, β-glucuronidase; GNS, *N*-acetylglucosamine-6-sulfatase. This scheme was created using BioRender.com (license no. YA2720J3QO).

**Figure 2 diagnostics-14-01734-f002:**
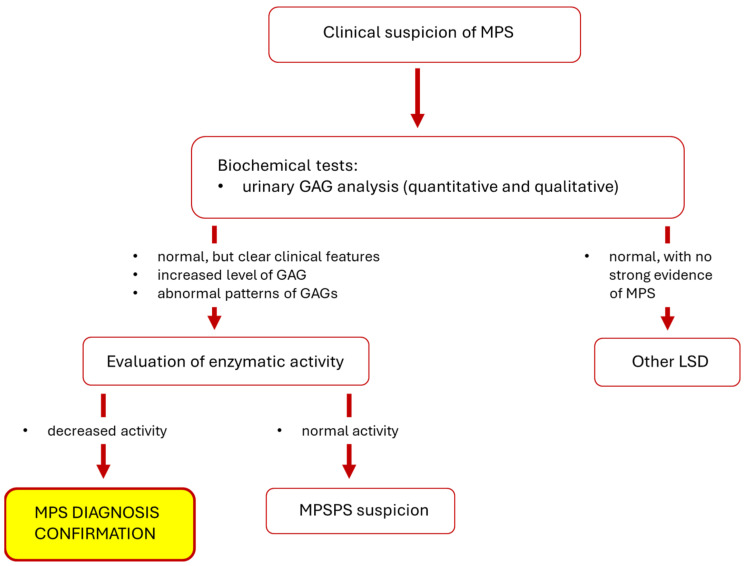
Simplified diagnostic scheme for MPS. Abbreviations: GAG, glycosaminoglycan; LSD, lysosomal storage disease; MPS, mucopolysaccharidosis; MPSPS, mucopolysaccharidosis-plus syndrome.

**Table 1 diagnostics-14-01734-t001:** Characteristics of classical/conventional types of MPS ^a^.

MPS Type	Defective Gene	Deficient Enzyme	Stored GAG ^b^	Neurological Symptoms ^c^
MPS I	*IDUA*	α-L-iduronidase	HS, DS	Impaired cognitive function, language, and speech abilities, behavioral abnormalities (excessive silencing), sleeping problems, and/or epileptic seizures
MPS II	*IDS*	Iduronate-2-sulfatase	HS, DS	Developmental delay, mental retardation, and behavior problems (aggression, over-excitability)
MPS IIIA	*SGSH*	Heparan-*N*-sulfatase	HS	Developmental delay, cognitive impairment, behavioral disorders (impulsivity, aggression, anxiety disorders, autistic behavior), sleeping problems
MPS IIIB	*NAGLU*	α-*N*-acetylglucosaminidase		
MPS IIIC	*HGSNAT*	Heparan α-Glucosaminide *N*-acetyltransferase		
MPS IIID	*GNS*	*N*-Acetylglucosamine-6-sulfatase		
MPS IVA	*GLANS*	*N*-Acetylglucosamine- 6-sulfate sulfatase	C6S, KS	Absence or mild neurological disorders as a consequence of secondary disturbances
MPS IVB	*GLB1*	β-Galactosidase	KS	
MPS VI	*ARSB*	*N*-acetylglucosamine-4-sulfatase (arylsulfatase B)	DS, C4S	None
MPS VII	*GUSB*	β-Glucuronidase	HS, DS, C4S, C6S	Impaired cognitive, language, and speech abilities, behavioral abnormalities, sleep problems, and/or epileptic seizures
MPS IX	*HYAL1*	Hyaluronidase	Hyaluronan	None
MPS X	*ARSK*	Arylsulfatase K	DS	None

^a^ MPS types in which typical features of MPS (an enzyme defect and increased GAG levels) are observed simultaneously; ^b^ Abbreviations: C4S, chondroitin 4,6-sulfate; C6S, chondroitin 6-sulfate; DS, dermatan sulfate; GAG, glycosaminoglycan; HS, heparan sulfate; KS, keratan sulfate, ^c^ central nervous system (CNS) disturbances can occur in non-neuronopathic types but they are caused by secondary and/or tertiary pathological changes in cells, and their incidence is not high.

**Table 2 diagnostics-14-01734-t002:** Symptoms that may indicate a need for diagnosis in the direction of MPS.

Typical MPS		Mild/Attenuated MPS	
Somatic thickened facial features recurrent infections of the upper respiratory tract, lower than average height, joint stiffness/excessive mobility abnormal dentition hepatosplenomegaly heart abnormalitie	Neurological developmental delay regression of skills behavioral disorders	Somatic corneal clouding retinal degeneration joint contractures without inflammatory signs carpal tunnel syndrome, and bone deformities tracheomalacia and/or stenosis hypoacusia hernias	Neurological sleep disturbances

**Table 3 diagnostics-14-01734-t003:** Summary of arguments for and against the occurrence of MPS IIIE in humans.

Why YES?	Why NOT?
Deficiency in activity of ARSG, an enzyme involved in HS degradation. The ARSG-deficient mouse model shows a similarity of symptoms to mouse models of other MPS III. In addition to phenotypes with distinct features and symptoms characteristic of MPS, there are mild/attenuated forms of this disease in which the disorders develop later and the characteristic changes are either mild or do not occur at all. Even in the ‘classic’ MPS III disease, the age of onset of some symptoms may vary depending on the subtype.	Symptoms of the *Arsg* KO mouse model do not correspond to those of patients with homozygous *ARSG* defective variants, in contrast to mouse models of other MPS III subtypes. In humans, the variant in the *ARSG* gene has already been described as USH type IV, with symptoms characteristic of this disease. Patients with *ARSG* pathogenic variants do not present symptoms of MPS, including CNS abnormalities typical of MPS III. The mouse model of *Arsk* deficiency does not correspond to the symptoms observed in patients with *ARSK* deficiency. Although a classification of MPS IIB has been proposed, patients are currently diagnosed with MPS X. Despite a considerably greater similarity of symptoms, MPSPS is not commonly classified as a type of MPS. Examples where a variant in the same gene causes different diseases are not uncommon, and sometimes involve genes that cause MPS, e.g., MPS IVB and GM1 gangliosidosis or MPS IIIC and RP. It is commonly accepted that deficiency in activity of a specific enzyme involved in GAG degradation and accumulation of particular GAG(s) are necessary for the diagnosis of MPS; if abnormal enzyme activity does not coexist with other biochemical abnormalities/clinical features, it is not sufficient to make a diagnosis of MPS.

## Data Availability

Not applicable.

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
