# Peer review of "Mucopolysaccharidosis Type IIIE: A Real Human Disease or a Diagnostic Pitfall?"

_diagnostics, 2024, doi:10.3390/diagnostics14161734_

Round 1

Reviewer 1 Report

Comments and Suggestions for Authors

The authors show an appreciation for the mucopolysaccharidoses (MPS) and the varied clinical findings and laboratory diagnostic approaches and pitfalls.  The topic is fun to read in this world of adding new diagnoses based upon gene variants, or certain phenotypic findings.  The discussion of pleiotropy was stimulating, as was the comparison between the severity levels of the two main types of MPS conditions.  The diagnostic algorithm was helpful too.  They make a convincing case for not creating a new MPS yet, but they still open the door to that happening.

Comments on the Quality of English Language

I'm not sure if the journal accepts the word "mutation," but most people use the word "variant," and the word mutation occurs in many places in the paper.  In one place it says there are 12 types of MPS and this would be the 13th, but in another, it says it would be the 14th --line 41.  Discussions could be tightened up more--there is a fair amount of back and forth, "on the other hand" type of discourse and repetition in the discussion, and I had to read it twice to be sure I understood it.

Author Response

COMMENT 1: I'm not sure if the journal accepts the word "mutation," but most people use the word "variant," and the word mutation occurs in many places in the paper. 

RESPONSE 1: According to reviewer’s recommendation, the term “mutation” has been replaced by “variant” throughout the text.

COMMENT 2: In one place it says there are 12 types of MPS and this would be the 13th, but in another, it says it would be the 14th --line 41. 

RESPONSE 2: We thank the reviewer for this note. “The 14th“ was a typographical error – it should be “the 13th”. This was corrected in the revised manuscript.

COMMENT 3: Discussions could be tightened up more--there is a fair amount of back and forth, "on the other hand" type of discourse and repetition in the discussion, and I had to read it twice to be sure I understood it.

RESPONSE 3: The text has been modified according to the reviewer’s recommendations.

Reviewer 2 Report

Comments and Suggestions for Authors

Dear Editor many thanks for asking me to review this paper. I congratulate the authors for addressing this topic. Some of the topics were out of my knowledge and I felt that I cannot comment but this overall this is a very good read. 

This is a very interesting article evoking new thoughts and review of our current understanding of the MPS disease. The topic of weather to consider MPSIII has been very well appraised with supportive literature review.

After reading the paper I have felt that I have learnt something new today. The English language is very professional and easy to read.

I cannot seem to find any issues to comment , I cannot seem to make any suggestions . 

Some minor issues 

Figure 2: Please elaborate the words below for the benefit of reader at every level - MPS, MPSPS, GAG's, LSD 

The authors have taken a balanced approach and concluded that we do not have enough evidence to call this MPS type IIIE. I feel this is a very important message to take home after reading the paper. 

Author Response

COMMENT 1: Figure 2: Please elaborate the words below for the benefit of reader at every level - MPS, MPSPS, GAG's, LSD 

RESPONSE 1: As requested by the reviewer, the abbreviations are explained in the legend to Figure 2.

COMMENT 2: The authors have taken a balanced approach and concluded that we do not have enough evidence to call this MPS type IIIE. I feel this is a very important message to take home after reading the paper. 

RESPONSE 2: We thank the reviewer for this comment. This is what we actually wanted to tell.